# Prediction of edapho-climatic parameters in the incidence of *Campylobacter* spp. in northwestern Mexico

**Yasiri Flores[1]☯, Andrea Chaves[1,2]☯, Gerardo Suzán[1]***

**1** Facultad de Medicina Veterinaria y Zootecnia, Universidad Nacional Autónoma de México, México City, México, **2** Escuela de Biología, Universidad de Costa Rica, San José, Costa Rica

☯ These authors contributed equally to this work.
* gerardosuz@gmail.com

**Data Availability Statement:** The data that support the findings of this study are available in a Supporting Information file (S1 Data).

**Funding:** This work was supported by the CONACyt project, Frontiers of Science 2016 (No.

## Abstract

*Campylobacter* spp. is one of the main causes of enteric zoonotic infections worldwide. In Mexico, although a commonly detected pathogen in both children and adults, there is limited surveillance and few studies. The northern part of Mexico stands out for an unnoticed outbreak of *Campylobacter jejuni* due to contaminated drinking water, which caused an abrupt increase in Guillain-Barré syndrome in the local population. Although it is suggested that its distribution in nature is related to edaphic and climatic factors, this relationship is scarcely known. To understand abiotic factors driving the occurrence and prevalence of *Campylobacter* spp. in three municipalities from three states in northwestern Mexico (Chihuahua, Sonora, and Baja California), we used the kriging interpolation method of unsampled areas and the correspondence analysis of 23 environmental variables. Of the three municipalities evaluated, Janos in Chihuahua (CHIH), has the highest number of geographic areas classified as high and medium incidence, followed by Santa Cruz, Sonora (SON) and Mexicali, Baja California (BC). Mexicali (BC) edaphic variables limit the potential incidence of the bacterium, mainly due to the lack of soil moisture and its difficulty of surviving on dry surfaces, related to electrical conductivity and salinity. Janos (CHIH) presents limitations in terms of soil water availability, although its presence is more heterogeneous (2 to 8 months). Santa Cruz (SON) has the highest soil water availability (4 to 5 months), and presents pH, texture and low percentage of salinity conditions for the potential incidence of *Campylobacter* spp. Mexicali (BC) reports a temperature in the warmest month of up to 43˚C, which could influence the presence of thermophilic species. The annual precipitation is another limiting factor for the potential incidence of *Campylobacter* spp. since it does not exceed 509.5 mm, contributing to Janos (CHIH) as the municipality with the highest potential incidence of this bacterium.

## Introduction

*Campylobacter spp.* -curved Gram-negative bacillus-, is a leading cause of foodborne bacterial zoonotic infections worldwide [1]. It is considered by the World Health Organization (WHO)

2016-01-1851 to GS; and postdoctoral fellowship to AC) and the DGAPA postdoctoral fellowship program (YF). The funders had no role in study design, data collection and analysis, decision to publish, or preparation of the manuscript.

**Competing interests:** The authors have declared that no competing interests exist.

as the primary etiological agent of diarrhea in humans in developed countries and the second or third in developing countries, including countries in Latin America [2, 3]. In Mexico, reports of campylobacteriosis in animals are scarce and the public health impact is unknown. Isolated investigations have reported the presence of *Campylobacter* spp. in humans, food, and food-animals [4]. Species reported include *Campylobacter jejuni*, *Campylobacter coli* and *Campylobacter fetus* [5–7]. *Campylobacter* spp. has been found in asymptomatic children and in children with diarrhea [4, 8]. In addition, in Mexico Guillain-Barré syndrome is the most frequent cause of paralysis in children under 15 years of age and has been correlated with previous *C. jejuni* infection in up to 40% of cases [8]. In June 2011, a cluster of suspected cases were reported in the northern border of Mexico with the USA due to the presence of *C. jejuni* in water used for human consumption [9].

The classification of the genus has been subject to revisions, with some authors reporting up to 25 different species [10]. However, 17 species and 6 subspecies are currently accepted [2], of which three species are recognized for their medical and veterinary importance: *C. jejuni*, *Campylobacter coli* and *Campylobacter lari*. It is associated with the digestive tract of mammals and birds (domestic and wild), and a wide variety of wild and domestic animals are recognized as natural reservoirs including poultry, cattle, sheep, pigs, rodents, dogs and cats [1, 11–13]. However, the actual number of wild species that act as reservoirs is still unknown [14, 15].

Soil, manure, aquatic environments and water are the natural niches where *Campylobacter* spp. are found and transmission to humans and susceptible animals can occur [16]. The growth and survival of *Campylobacter* spp. is related to external environmental factors such as water, heat, UV radiation and salinity [17]. If we consider that many of its reservoir groups, especially mammals, use or live in and within the soil and contribute directly to its formation, it is essential to take into account the influence of edaphic characteristics in the maintenance of *Campylobacter spp.* [18]. For example, rodents and carnivores that dig deep into the soil carry out a considerable mixing of the soil, often bringing subsoil to the surface and leaving a cavity into which the top soil can fall and accumulate in the subsoil [15, 19]. In addition to these edaphic factors, the dynamics of seasonal change during the year and modifications over long periods, both related to climatic factors, are associated to the presence of *Campylobacter spp.*, variables that can help predict its potential occurrence [20, 21] and establish regions of high, medium or low risk of infection.

*Campylobacter* can survive for long periods in sources of water and soil, especially during the winter [22]. According to Nicholson et al. [23], after application of *Campylobacter* contaminated manure to soil, the bacteria survived in the soil for up to one month after application, both in the sandy soils of arable crops and in the clay soils of grassland. Additionally, Jaderlund et al. [24] determined that regardless of the inoculation dose of *C. jejuni*, and the strategy employed, the *C. jejuni* content in the soil remained relatively constant throughout the study period (21 days). Because it is sensitive to ultraviolet light and higher temperatures during the summer, it is rapidly decimated [21, 22]. However, animals acting as reservoirs are involved in the recontamination of environmental sources, which is worth monitoring in regions dedicated to agricultural production and with a variety of climates [25, 26]. Northwest Mexico ranks fourth and fifth in food production in Mexico, Chihuahua (CHIH) and Sonora (SON), respectively, with extensive agricultural lands, while Baja California (BC) ranks second in the country in productive value per hectare harvested [27].

The climates in northwestern Mexico are very extreme; temperatures show a wide average oscillation. The difference between the average temperature of the hottest month and the coldest month of the year can be as much as 20°C in northern CHIH and Coahuila, and the oscillation between average daytime temperatures can be as much as 24°C. Average annual

precipitation ranges from 150 to 2000 mm [28]. This diversity of characteristics highlights the relevance of understanding the presence of *Campylobacter* in northwestern Mexico by considering environmental factors, including edaphic-climatic characteristics that provide a more complete picture identifying high risk areas and developing prevention strategies.

Habitat suitability for different species occurrence in time and space, including the genera *Campylobacter spp*. can be projected with species distribution models. Consisting of the mathematical or statistical relationship between the actual known distribution of the species and a set of independent variables that are used as indicators [29, 30]. One of the most widely applied local spatial interpolation techniques is the geostatistical or kriging method, which incorporates a mathematical model to describe the spatial variation of the data through a measure of the spatial autocorrelation between pairs of points, which describe the variance at a given distance [31]. Currently, in these models we can include environmental, biological, and anthropogenic variables, which makes it easier to decide priorities [32, 33].

The kriging method allow us to use environmental and edaphic variables associated with a previously determined *Campylobacter spp*. presence, hence we can establish the degree of spatial autocorrelation between sampling sites to obtain estimates in unmeasured sites. With kriging we can establish the degree of spatial autocorrelation between sampling sites to obtain estimates in unmeasured sites, since it associates the Best Linear Unbiased Predictor (MPLI) term and more adequate, minimizing the error variance in the prediction. It is based on the fact that natural variables are generally continuously distributed [34, 35]. Providing a measure of the error or uncertainty of the estimated surface, therefore, a theoretical distribution can be associated to each point of the estimated space, which also allows the possibility of performing probabilistic simulations, and showing the result as the probability that the variable reaches a certain value [31, 36, 37].

Therefore, using kriging method, the objective of this research was to determine the importance of the sources of infection and other elements such as species studied, country, climatic season, and year in the prevalence value; to define which edaphic and climatic factors influence the occurrence and prevalence of *Campylobacter spp*. in Northwest Mexico.

## Methods

### Area of study

The study was carried out in three municipalities in the northwestern region of Mexico: 1) Mexicali in BC, 32° 43' and 30° 25' north latitude and 114° 42' and 115° 56' west longitude, altitude of 3 masl. 2) Santa Cruz in SON, 31°13' north latitude and 110°35' west longitude, altitude of 1,463 masl. 3) Janos in CHIH, north latitude 39°53' and 108°44' west longitude, altitude 1,380 masl (Fig 1).

### Data collection and calculation of bioclimatic profile variables for northwestern Mexico

Data corresponding to edaphic variables were obtained from 170 samples of the soil profile dataset at scale 1:250 000, series II (National Continuous). Serie II: 2002–2007 [27]: pH, texture, moisture percentage and salinity, the latter measured by electrical conductivity. The climatic data were obtained from 23 meteorological stations present in the three municipalities and adjacent ones from the National Meteorological Service and the corresponding calculations were made for the 19 variables that make up the bioclimatic profile (Table 1) [38, 39].

A systematic search of world scientific literature was carried out in which edaphic, meteorological, water bodies, vegetation and land use factors associated with the presence of

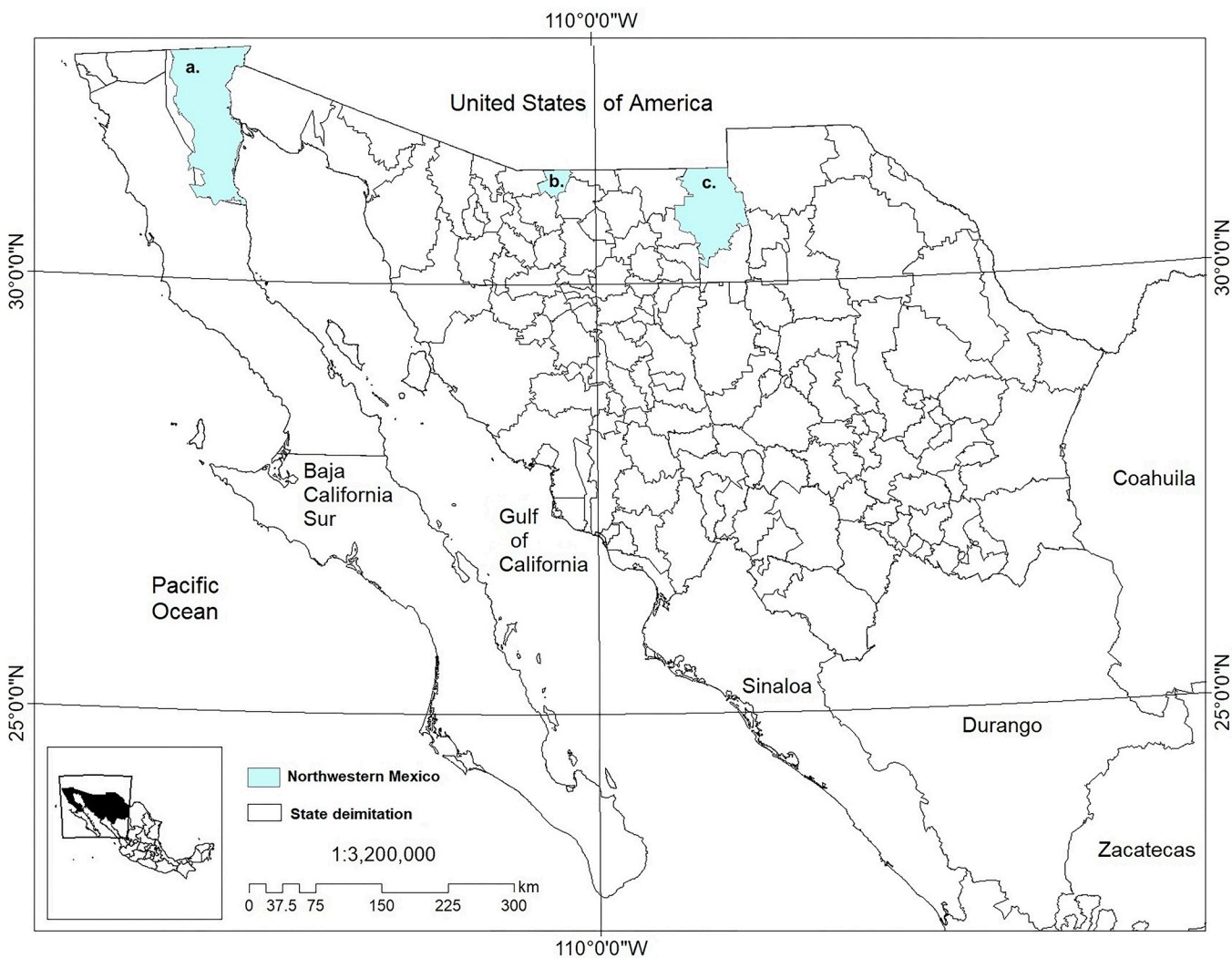

**Fig 1.** Demarcation of the area of study in a. Mexicali (Baja California), b. Santa Cruz (Sonora) and c. Janos (Chihuahua), Mexico. This map was elaborated in-house with ArcGIS 10.8.2 program from a standard layer of the database of National Institute of Statistics and Geography (INEGI) of Mexico, a free and open-source database (https://inegi.org.mx/app/mapas//).

*Campylobacter* spp. in soil were established. At the same time, epidemiological records of *Campylobacter* spp. were consulted, without success, from Mexican public health and agricultural authorities and epidemiological bulletins. According to what was consulted in these institutions and corroborated by the systematic search carried out, it was not possible to obtain data for the northwestern region of Mexico that would be useful for our study.

## Generation of thematic mapping by kriging interpolation and modeling of the potential distribution of *Campylobacter* spp. for northwestern Mexico

For edaphic and climatic variables, ordinary kriging interpolation was performed: this produces reliable restructured surfaces by considering the spatial structure in the raw variables to minimize error variance [40–42]. In this geostatistical design, an ordinary predictive kriging was worked, and a spherical model was fitted to the spatial structure, up to a distance above which the autocorrelation is zero, in this case 15 meters from any sample point. The

**Table 1. Climatic data obtained from 23 meteorological stations present in the three municipalities and adjacent ones from the National Meteorological Service and the corresponding calculations made for the 19 bioclimatic profiles in Mexicali (Baja California), Santa Cruz (Sonora) and Janos (Chihuahua), Mexico.**

| Variable | Mexicali, Baja California | Santa Cruz, Sonora | Janos, Chihuahua |
|---|---|---|---|
| BIO1 Average annual temperature | 24.4–15.1 | 17.7–16.7 | 26.1–25.6 |
| BIO2 Mean diurnal temperature range (temp max—temp min; monthly average) | 20.5–12.4 | 18.6–18.0 | 17.9–16.8 |
| BIO 3 Isothermality (BIO1/BIO7)*100 | 56.0–45.4 | 53.8–53.2 | 51.9–48.0 |
| BIO4 Seasonality of temperature (coefficient of variation) | 783.5–142.8 | 6.42–6.31 | 628.7–51.0 |
| BIO5 Maximum temperature of the warmest month | 43.8–27.0 | 36.8–36.1 | 34.2–33.6 |
| BIO6 Minimum temperature of the coldest month | 8.4–1.3 | 1.55–1.01 | 0.0 –-1.0 |
| BIO7 Annual temperature range (BIO5-BIO6) | 39.9–19.7 | 34.6–33.7 | 34.3–33.83 |
| BIO8 Average temperature of the wettest quarter | 33.6–21.8 | 24.8–23.6 | 23.9–22.5 |
| BIO9 Average temperature of the driest quarter | 18.6–13.5 | 13.7–13.0 | 14.0–13.5 |
| BIO10 Average temperature of the warmest quarter | 27.3–15.7 | 21.6–20.8 | 21.0–20.78 |
| BIO11 Average temperature of the coldest quarter | 14.5–12.0 | 10.7–9.4 | 21.0–20.78 |
| BIO12 Annual precipitation | 271.6–32.5 | 513.7–503.4 | 467.1–463.0 |
| BIO13 Precipitation of the rainiest month | 64.5–6.7 | 127.6–115.9 | 113.8–103.1 |
| BIO14 Precipitation of the driest month | 1.4–0.0 | 8.4–5.8 | 8.2–6.8 |
| BIO15 Seasonality of precipitation (coefficient of variation) | 115.4–11.0 | 7.64–7.40 | 94.4–85.5 |
| BIO16 Precipitation of the rainiest quarter | 77.9–6.9 | 288.2–283.4 | 278.4–258.6 |
| BIO17 Precipitation of the driest quarter | 130.3–4.7 | 74.3–52.8 | 33.8–27.1 |
| BIO18 Precipitation of the hottest quarter | 23.5–0.1 | 36.1–31.0 | 65.6–56.7 |
| BIO19 Precipitation of the coldest quarter | 146.2–7.7 | 109.9–83.5 | 73.9–42.9 |

biophysical profiles thus obtained synthesize the environmental conditions of the sites analyzed and represent the factors that make up the potential distribution area of *Campylobacter* spp.

Kriging uses the degree of spatial autocorrelation between sampling sites to obtain estimates in unmeasured sites, associates the term Best Linear Unbiased Predictor (MPLI) and is the most appropriate, in the sense that it minimizes the error variance in the prediction; it is because natural variables are generally distributed continuously [34, 35]. According to Burrough and McDonell [34], the adoption of the stochastic approximation implies that, at any point in space, there is not a single value of an attribute but a set of values. The value observed at a point is the result of a random process, with a specific probability distribution. This assumes that at all points in space there is variation, so that a regionalized variable is associated with spatial location. From a mathematical point of view, a regionalized variable is simply a function f(x), which has a certain value for all x-coordinates in space. The regionalized variable theory assumes that the spatial variation of any attribute can be expressed by the sum of three components:

- Structural component, which has a constant mean or trend.

- Random, but spatially related component, known as variation of the regionalized variable

- Residual error, a spatially uncorrelated random component.

The value of the random variable Z at point x is expressed as follows:

$$Z(x) = m(x) + \varepsilon'(x) + \varepsilon''$$

Where m(x) is a function describing the structural component, the function ε'(x) represents the spatially correlated residuals of m(x) (i.e., the regionalized variable), and ε"(x) is the residual error that has a normal distribution with mean 0 and variance σ2.

From the records of the presence of *Campylobacter spp*. the selected factors and the cartography corresponding to water bodies, vegetation and land use, roads and human settlements and Natural Protected Areas (NPAs), topological superimpositions were carried out to extract the values of each site in the form of a raster. Subsequently, cartographic algebra was carried out, which consists of obtaining new layers of information for which a set of calculation tools is available with data matrices that receive the generic name of map algebra and includes a large set of operators or algorithms executable on one or more various input raster layers to produce one or more output raster layers [43]. For this purpose, different investigations were considered in which they report the behavior of the bacteria and their relationship with multiple factors, which epidemiologically have been classified into three large groups according to their origin: environmental factors, reservoirs, and risk factors (Table 2).

**Table 2. Environmental factors, reservoirs and risk factors associated with the presence of Campylobacter.** With information from: [10–13, 16, 63, 68, 76, 80, 83, 86–106].

| Environmental factors | Reservoirs | Risk Factors |
|---|---|---|
| **Soil characteristics:** Type of agricultural soils especially when irrigated with treated water. Improper handling of manure used in planting can be a risk factor contributing to food contamination. The species can remain viable 9–32 days in manure. **Texture** has the capacity to remain viable for up to 8 days in sandy and grassy soils, and up to 32 days in clay soils. pH The optimum pH is 6.5 to 7.5, minimum of 4.9 and maximum of 8. **Salinity** Salinity causes a decrease in water availability. Saline soils are those that contain significant amounts of salts more soluble than gypsum. Salinity is measured by electrical conductivity. **Moisture** highly sensitive to moisture loss and does not survive well on dry surfaces. | It is widely distributed in nature and has demonstrated considerable ecological diversity: soils, anaerobic sludge, manure, untreated or contaminated water, as well as in roots of certain plants and vegetables harvested from contaminated soil or irrigated/washed in contaminated water. A wide variety of domestic animals are its natural reservoir, such as cattle, pigs, sheep, poultry, goats, dogs and cats. Although wild animals are carriers and have a potential role in the transmission of thermophilic Campylobacter, their relative importance is less than that of domestic animals. In general, in wild animals, prevalences vary depending on the ecology of each species, including feeding habits, coprophagy, migration patterns, habitat, breeding or moulting period. In wildlife it has been reported in reptiles, rodents, monkeys, migratory birds such as ducks, geese, gulls, terns, pigeons, swallows, hawks, owls and crows. While flies and beetles can carry the bacteria in their exoskeleton and may serve as vectors. | **Associated with human activity.** Farmers and personnel visiting farms represent a risk factor. Campylobacter has been isolated from the clothing, hands and footwear of farm personnel. **Associated with farms.** To consider bedding, feed, drinking water. The most frequent insects on farms are flies, beetles and cockroaches. **Associated with handling.** Wild animals, including rodents, as well as other domestic animals on or near the farm. In addition, disposal of waste and carcasses is important. Finally, the rinsing of lots because contaminated cages in one farm and improperly cleaned, introduce the infection in the next farm. |
| **Seasonality.** An important feature of the epidemiology of thermophilic Campylobacter is its marked seasonality. In temperate climates the incidence of campylobacteriosis has a seasonal cycle, with a peak in summer in northern hemisphere countries. However, the causes of this seasonal cycle are not clear. Recent studies have shown a correlation between environmental temperature and the number of **Campylobacter cases.** The risk associated with rainfall would increase in temperate climates that are associated with increased activity of flies, which can act as vectors of this bacterial enteropathogen. **Temperature.** The optimum range is 37 to 43˚C, the maximum 45˚C and the minimum 30˚C. However, it depends on the species since some can be distributed up to 25˚C and have been reported down to 13˚C. **Precipitation.** Abundant rainfall or floods have an influence, which can affect the frequency and level of contamination of drinking water. | **Bodies of water** They can be natural or artificial and are influenced by contamination with dead and decomposing animals, fecal material, and waste. Generally, occurs where there is use of untreated groundwater together with inadequate treatment of collected groundwater and contamination of distribution systems. | Transmission **Food—person** through consumption of food contaminated with the bacterium. **Animal/person—person** through fecal-oral route from infected animals on the farm or from pets, as well as from people already infected. **Food/water-soil-food** through cross-contamination on farms, in food processing, and in food preparation and cooking at home. |

## Results

### Data collection and calculation of bioclimatic profile variables for northwestern Mexico

The bioclimatic parameters for the surface of the three municipalities of Northwest Mexico: Mexicali (BC), Santa Cruz (SON) and Janos (CHIH) were calculated by station from the 1981–2010 monthly meteorological database of the National Meteorological Service, following the ANUCLIM methodology also used in Worldclim (Figs 2–6, S1 Data) [38, 39]. The superposition of these thematic layers shows the differences in environmental parameters needed in potential incidence, mainly soil moisture, pH, salinity, texture, as well as presence of water bodies, vegetation and land use, and the integration of the 19 variables of the bioclimatic profile (Table 3).

### Generation of thematic mapping by kriging interpolation and modeling of the potential distribution for northwestern Mexico

Superposition of the thematic layers showing the differences in the environmental parameters needed in potential incidence mainly for soil moisture, pH, salinity, texture, presence of water bodies, vegetation and land use and the integration of the 19 variables of the bioclimatic profile (Table 3).

Potential areas of *Campylobacter spp*. incidence in Mexicali (BC), Santa Cruz (SON) and Janos (CHIH) were weighted considering a high, medium, and low level of *Campylobacter* spp. incidence. Of the three municipalities evaluated, Janos (CHIH) has the highest number of geographic areas classified as high and medium incidence, followed by Santa Cruz (SON) and lastly, Mexicali (BC) with a greater number of geographic areas with low incidence (Fig 7).

The high incidence sites for Janos (CHIH) coincide with agricultural areas with annual irrigation (permanent and semi-permanent), as well as chaparral and secondary shrubby oak forest vegetation. Santa Cruz (SON) coincides with agricultural areas with annual irrigation (permanent and semi-permanent) and human settlements. For Mexicali (BC) it is associated with desert chaparral and chaparral.

Considering the main parameters associated with each study region, the maximum incidence of *Campylobacter* spp. in the Janos (CHIH) region was related to the presence of a pH ranging from 6 to 8 and salinity from 0.2 to 1.9, in contrast to Mexicali (BC) which can be very high and in Santa Cruz (SON) extremely low for both parameters. For Janos (CHIH), the fine soil texture, associated with high incidence, is more homogeneous throughout the region, while in Mexicali (BC) it is associated with small patches. In Santa Cruz (SON), despite a homogeneous composition of fine soil texture, the other characteristics associated with high risk of *Campylobacter* spp. incidence are lacking. In Janos (CHIH), soil moisture content is more variable and ranges from 2 to 8 months, while in Mexicali (BC) most of the soil is dry and in Santa Cruz (SON) soil moisture is present only 4 to 5 months per year.

With respect to temperature, if we compare the three municipalities, in Janos (CHIH) the average temperature (B1) is the highest and ranges from 20 to 27˚C, the maximum temperature of the warmest month (B5) from 35 to 37˚C, the average temperature of the rainiest month (B8) from 24 to 27˚C and the average temperature of the warmest quarter (B10) is from 21 to 22.3˚C. In the case of annual precipitation (B12) Janos (CHIH) falls within a range of 267 to 458.7 mm which is more variable compared to Mexicali (BC) which ranges from 30.6 to 271.1 mm and Santa Cruz (SON) which ranges from 450.2 to 509.5 mm.

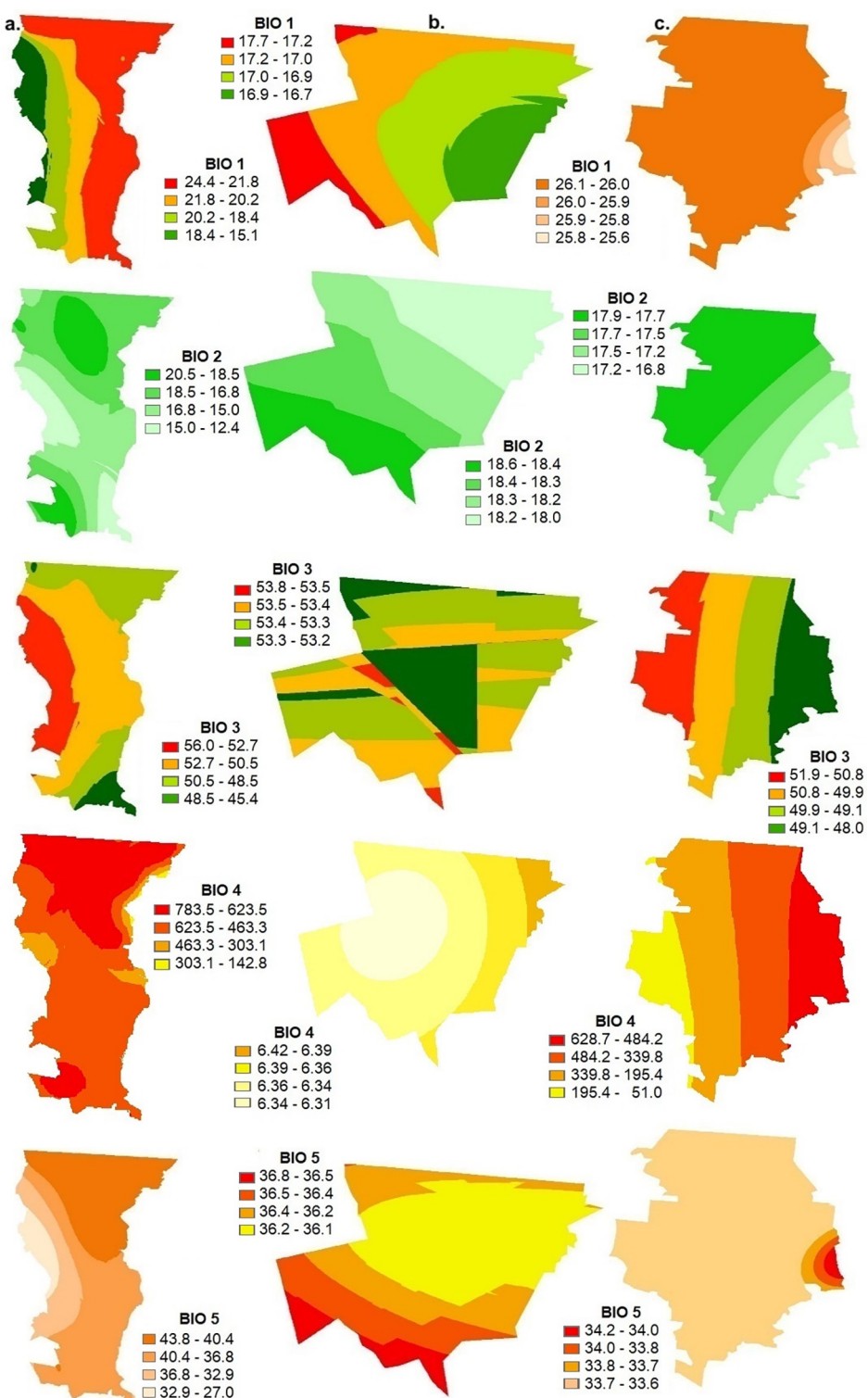

**Fig 2.** Geostatistical Kriging interpolation for the 19 variables that compose the bioclimatic profile in a. Mexicali (Baja California), b. Santa Cruz (Sonora) and c. Janos (Chihuahua). BIO1 mean annual temperature, BIO2 mean diurnal temperature range, BIO3 isothermality (index of temperature variability), BIO4 seasonality of temperature, BIO5 maximum temperature of the warmest month. This map was elaborated in-house with ArcGIS 10.8.2 program from a standard layer of the database of National Institute of Statistics and Geography (INEGI) of Mexico, a free and open-source database (https://inegi.org.mx/app/mapas//).

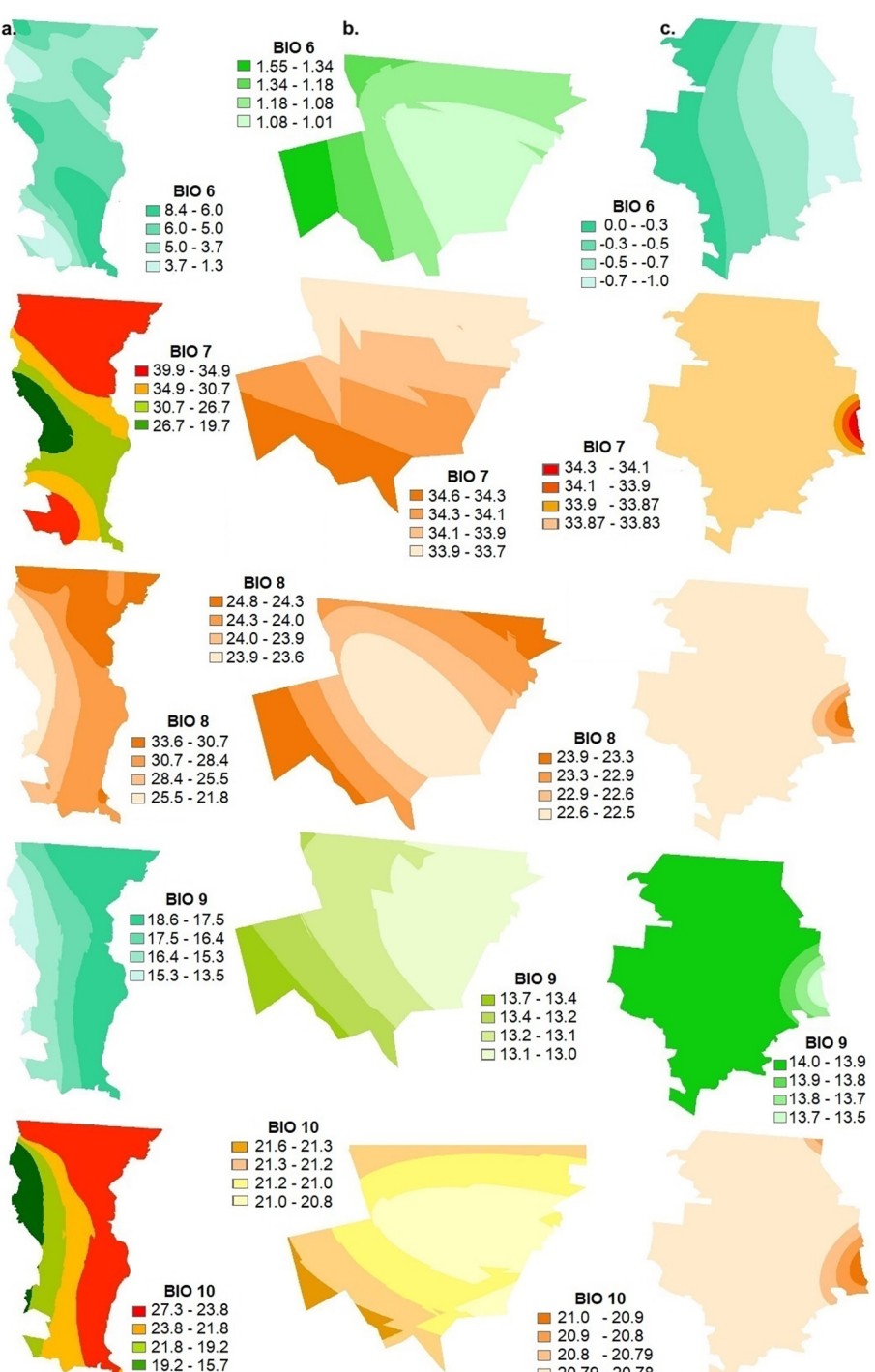

**Fig 3.** Geostatistical Kriging interpolation for the 19 variables that make up the bioclimatic profile in a. Mexicali (Baja California), b. Santa Cruz (Sonora) and c. Janos (Chihuahua). BIO6 minimum temperature of the coldest month, BIO7 annual temperature range, BIO8 average temperature of the rainiest quarter, BIO9 average temperature of the driest quarter, BIO10 average temperature of the warmest quarter. This map was elaborated in-house with ArcGIS 10.8.2 program from a standard layer of the database of National Institute of Statistics and Geography (INEGI) of Mexico, a free and open-source database (https://inegi.org.mx/app/mapas//).

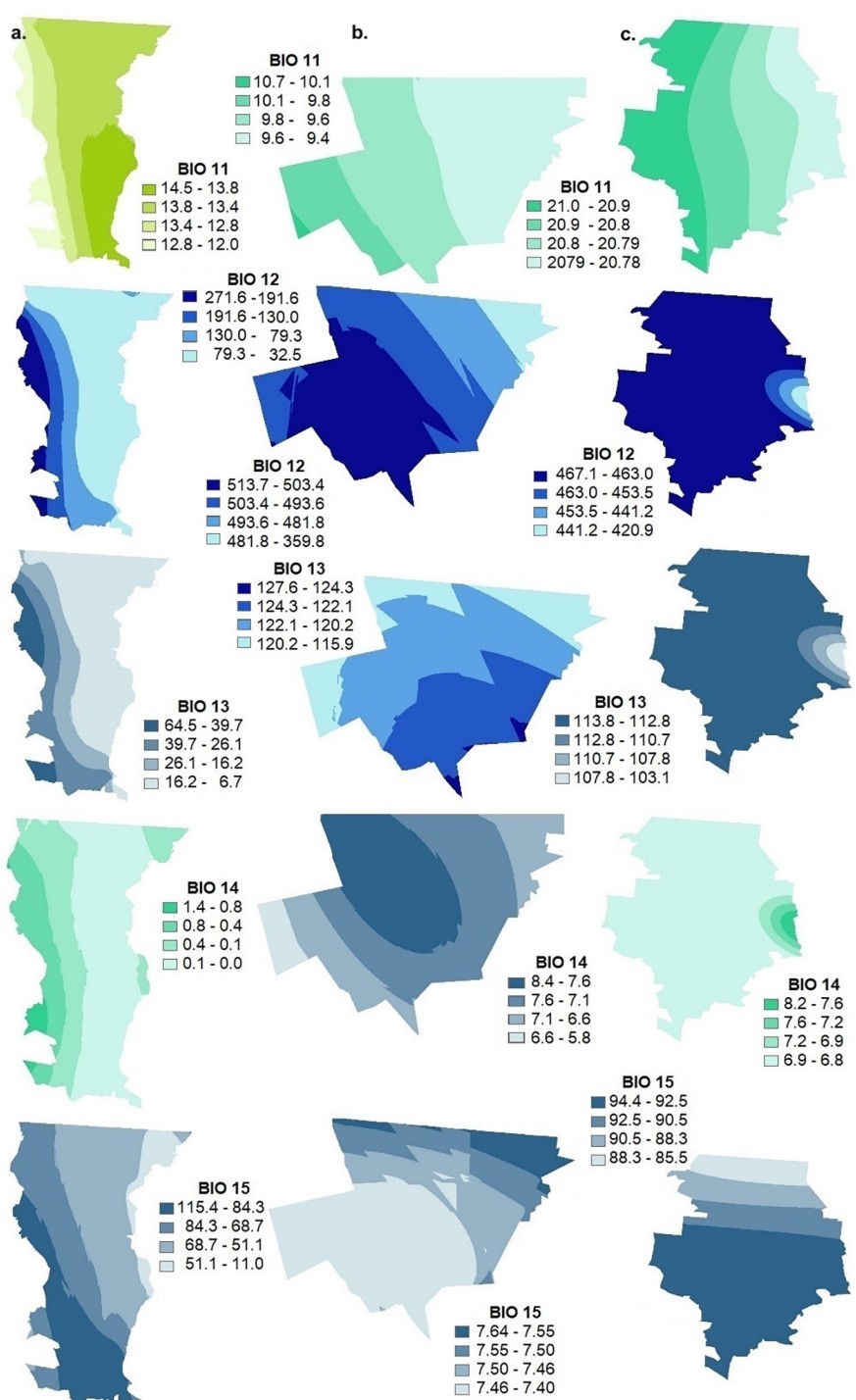

**Fig 4.** Geostatistical Kriging interpolation for the 19 variables that make up the bioclimatic profile in a. Mexicali (Baja California), b. Santa Cruz (Sonora) and c. Janos (Chihuahua). BIO11 average temperature of the coldest quarter, BIO12 annual precipitation, BIO13 precipitation of the rainiest month, BIO14 precipitation of the driest month. BIO15 seasonality of precipitation. This map was elaborated in-house with ArcGIS 10.8.2 program from a standard layer of the database of National Institute of Statistics and Geography (INEGI) of Mexico, a free and open-source database (https://inegi.org.mx/app/mapas//).

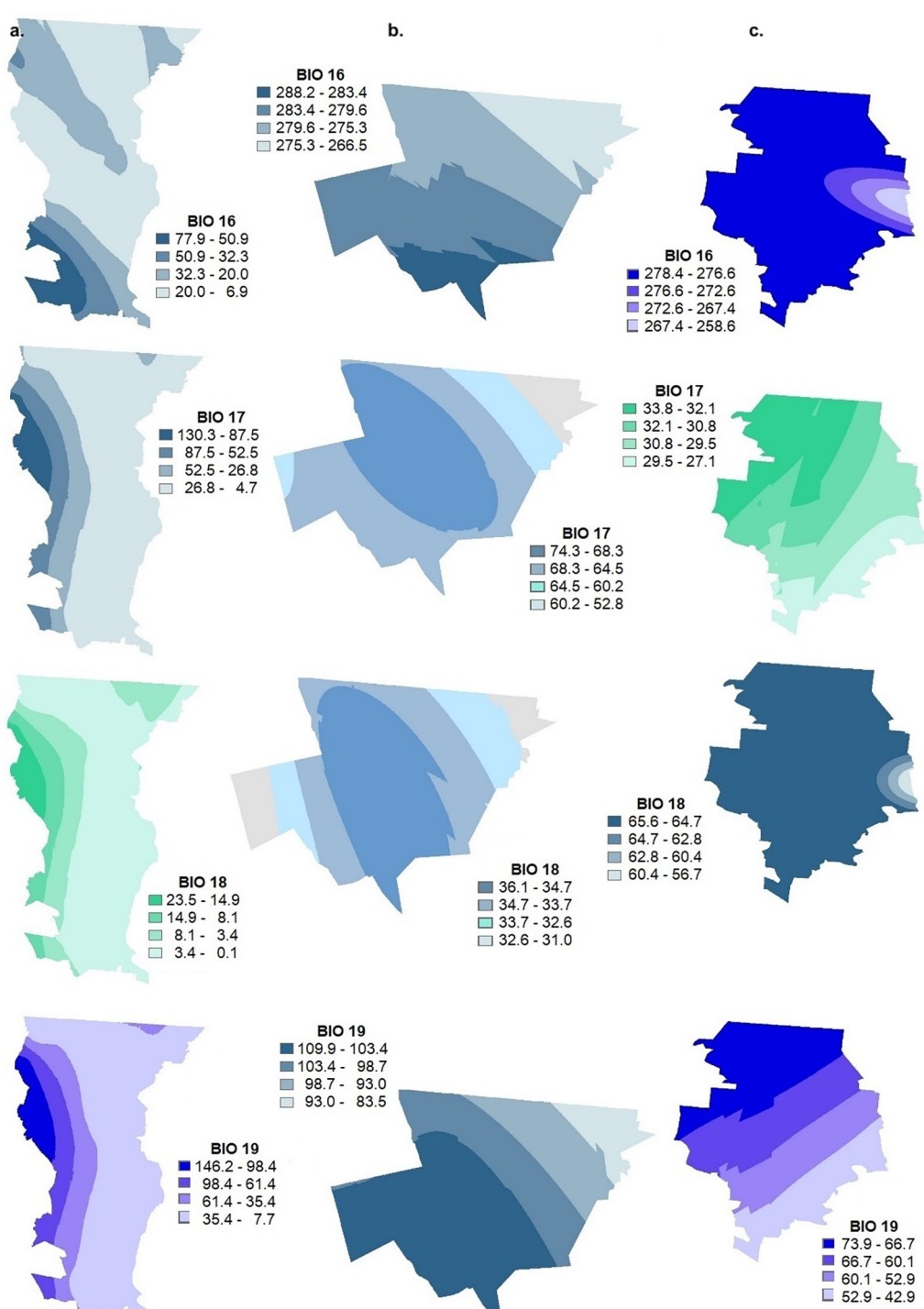

**Fig 5.** Geostatistical Kriging interpolation for the 19 variables that make up the bioclimatic profile in a. Mexicali (Baja California), b. Santa Cruz (Sonora) and c. Janos (Chihuahua). BIO16 precipitation of the rainiest quarter, BIO17 precipitation of the driest quarter, BIO18 precipitation of the warmest quarter, BIO19 precipitation of the coldest quarter. This map was elaborated in-house with ArcGIS 10.8.2 program from a standard layer of the database of National Institute of Statistics and Geography (INEGI) of Mexico, a free and open-source database (https://inegi.org. mx/app/mapas//).

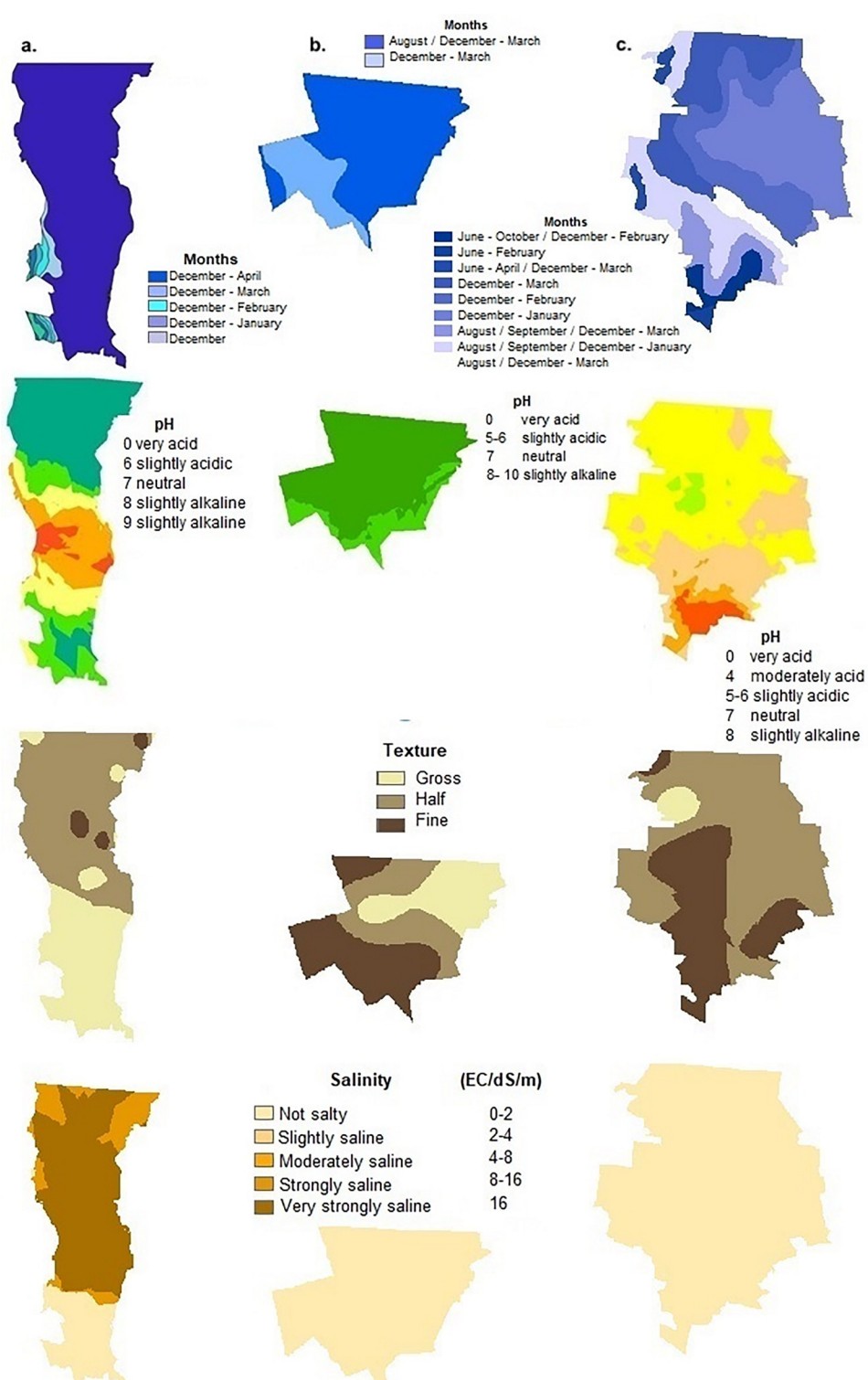

**Fig 6.** Geostatistical Kriging interpolation for edaphic variables: moisture, depth and pH in a. Mexicali (Baja California), b. Santa Cruz (Sonora) and c. Janos (Chihuahua). This map was elaborated in-house with ArcGIS 10.8.2 program from a standard layer of the database of National Institute of Statistics and Geography (INEGI) of Mexico, a free and open-source database (https://inegi.org.mx/app/mapas//).

**Table 3. Weighting of edaphoclimatic variables in the potential incidence of *Campylobacter spp.* for Mexicali (Baja California), Santa Cruz (Sonora) and Janos (Chihuahua).**

| Variable | Weighting of Potential Incidence |
|---|---|
| pH | High. 6.5 a 7.5<br>Medium. 4.9 a 6.5 y 7.5 a 8<br>Low. 8 a 9 |
| Texture | High. Fine (clayey) or its equivalent Ø < 0.002 mm<br>Medium. Coarse (sandy) or its equivalent 2 mm > Ø > 0.05 mm<br>Low. Medium (silty) or its equivalent 0.05 mm > Ø > 0.002 mm |
| Humidity | High. Acute with 365 days or its equivalent from six to 11 months. Undivided with 270 to 330 days<br>Medium. Ustic with 180 to 270 days or its equivalent from three to six months. Xeric with 90 to 180 days<br>Low. Aridic with 0 to ≤ 90 days or its equivalent of one to three months. |
| Salinity | High. 0 to 2 not saline, higher soil moisture content.<br>Medium. 2 to 4 slightly saline<br>Low. 4 to ≥ 16 moderately to very strongly saline lower soil moisture content. soil moisture content |
| Temperature | High. 30 to 45˚C<br>Medium. 25 to 30˚C<br>Low. 13 to 25˚C |
| Rainfall | High. Warm-humid (2000 to 4000 mm)<br>Temperate-humid (2000 to 4000 mm)<br>Warm-sub-humid (1000 to 2000 mm)<br>Medium. Temperate-sub-humid (600 to 1000 mm)<br>Low. Dry (300 to 600 mm)<br>Very dry (100 to 300 mm) |
| Vegetation and land and land use | High. Agricultural areas, human settlements and bodies of water.<br>Medium Secondary vegetation<br>Low. Forests, jungles, scrub, mesquite, mangrove, chaparral, tular, Grasslands, gallery vegetation, halophytes and sandy deserts. |

## Discussion

It is clear that any approach to investigating possible environmental factors associated with the increase in *Campylobacter* spp. cases must incorporate factors that may affect host exposure, including climatic conditions, soil types or conditions, as well as land use issues. Unlike other foodborne pathogens, *Campylobacter spp.* have generally been considered environmentally sensitive, fragile, and unable to multiply outside the host as a consequence of their thermophilic and microerophilic nature [16, 44, 45]. However, recent studies have shown it to be more resistant than previously thought [42] by reaching the viable but non-culturable cell (VBNC) stage and infecting susceptible hosts [11, 46]. Furthermore, the soil has been demonstrated to significantly contribute to the persistence of the bacterium in the environment. This is attributed to its relative significance as a potential infection source, linked to the bacterium's resistance to environmental conditions [18, 47]. With this study we intend to improve our understanding of the presence of *Campylobacter* spp. due to edaphic factors, presence of water bodies, vegetation and land use and the integration with meteorological factors that influence the distribution of *Campylobacter* spp. in the environment moderated by the combination of factors mentioned above.

We found that the two municipalities (Janos and Santa Cruz) with the highest risk zones for the presence of *Campylobacter* spp. agricultural soils with annual irrigation had a relationship with high incidence of the bacterium. It has been shown that landscapes with a larger agricultural extension with permanent irrigation have a higher environmental contamination of *Campylobacter* spp. [48]. In addition, meteorological conditions in Janos (CHIH) provide higher soil moisture and less extreme temperatures compared to the other two municipalities.

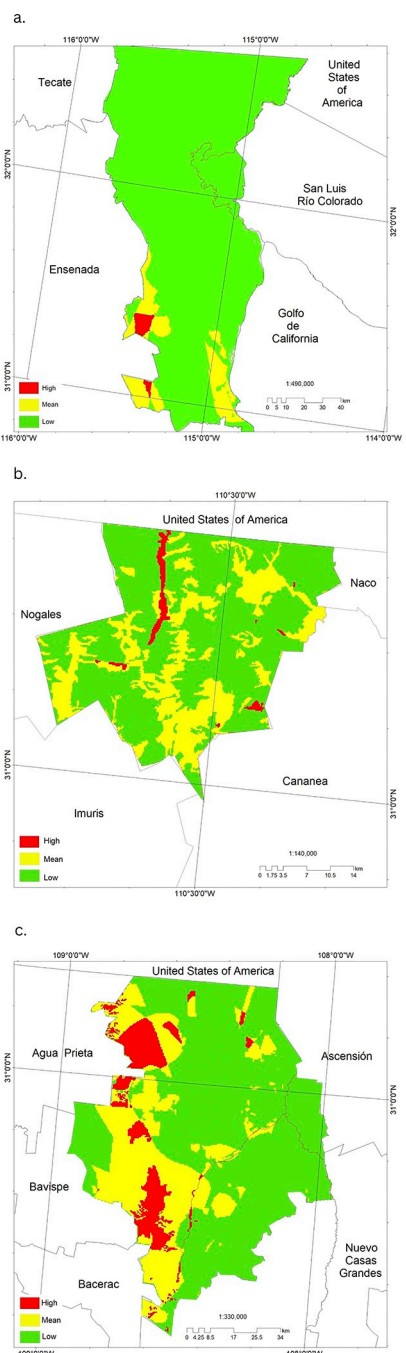

**Fig 7.** Potential incidence of Campylobacter spp. in the municipalities of a. Mexicali, Baja California, b. Santa Cruz, Sonora and c. Janos, Chihuahua considering edaphic variables (humidity, pH, salinity and texture), bioclimatic profile, presence of water bodies, vegetation and land use. This map was elaborated in-house with ArcGIS 10.8.2 program from a standard layer of the database of National Institute of Statistics and Geography (INEGI) of Mexico, a free and open-source database (https://inegi.org.mx/app/mapas//).

This may favor the presence and survival of *Campylobacter* spp. due to more stable temperatures and higher soil moisture [24]. Soil-borne pathogens survive better in a more open soil matrix that allows greater soil penetration and contains more organic matter, with less desiccation during dry weather. [49]. Although published data on *Campylobacter* spp. survival in

northern Mexico are not available, it would be expected that survival would be lower in regions with greater desiccation. Additionally, runoff is generally lower in less compacted soils [50], so the fine soil texture mostly found in Janos (CHIH) may favor *Campylobacter* spp. survival, favored by a pH and salinity suitable for their survival and therefore an increase in *Campylobacter* spp. exposure, especially those species known for their high survival in the environment such as *C. jejuni* [50].

Even though there is limited surveillance and few studies in Mexico, cases of campylobacteriosis in humans and especially in children are common. In a study [5] it was reported that 3% of asymptomatic children aged 3 to 7 years excreted *Campylobacter* spp. and in 15.7% of children with diarrhea *C. jejuni* was isolated [4]. The northern part of Mexico, specifically areas near San Luis Río Colorado, SON, Mexico and Yuma, Arizona, USA, have become of interest in the last decade. This was due to an outbreak of Guillain-Barré syndrome, where *Campylobacter* spp. exposure was established. It was suggested after an environmental assessment that the cases were due to a large *Campylobacter jejuni* outbreak that went unnoticed in this border area, related to a potential contamination of drinking water [9].

In Mexicali (BC), the municipality with the lowest potential incidence for *Campylobacter* spp., edaphic variables limit the potential incidence of the bacterium, mainly associated with the lack of soil moisture. The edaphic variables limit the potential incidence of the bacteria, mainly associated with the lack of moisture in the soil, favoring that the electrical conductivity and salinity exceed the moderately to very strongly saline range. Santa Cruz (SON) is the municipality, with an average potential incidence of *Campylobacter spp.* compared to the other two municipalities studied, with the highest water availability in 4 to 5 months per year and presents the other edaphic conditions for the potential incidence of *Campylobacter spp.* such as pH, texture, and low percentage of salinity. The municipality of Janos (CHIH), with a greater number of geographic areas with high and medium potential incidence of *Campylobacter* spp., also presents certain limitations in terms of water availability in the soil, however it is more heterogeneous, with availability from 2 to 8 months per year. (Table 2).

Survival in water varies among studies, for example Rollins et al. [51] reports a survival of 120 days, while Buswell et al. [52] reports a survival of 29 days. Furthermore, Nilsson et al. [53] evaluated the variation of survival in water depending on seasonality, and in the case of *C. jejuni*, a significantly better survival was observed in autumn than in spring. [51, 52]. In the study area watercourses are limited, mainly in the municipality of Santa Cruz (SON) and in general northern Mexico having dry to very dry landscapes all year round, it is common to use irrigation water for their agricultural activities. In addition, a common association has been demonstrated between irrigation water contaminated with *Campylobacter spp.* [48], which may favor the transmission of the bacteria to these soils. In addition to watercourses that may be contaminated by the presence of *Campylobacter spp.* infected feces from common domestic hosts in the area (e.g. cattle or poultry) or other domestic or wild hosts. Facilitating the presence of landscapes with possible propensity for transmission of the pathogen to other animals or even humans.

In accordance with most of the literature we report for Northwest Mexico a high variation in the prevalence range, which has been found in humans, domestic and wild birds [54–57] *C. jejuni* and *C. coli* also occur with the highest number of observations as reported in the literature in general. Both species related to species such as poultry, duck, turkey, cattle, sheep, wild birds and pigs [58, 59]. All reports that indicated animal contact as a source correspond to these two species.

There is a lack of information on *Campylobacter spp.* records in the Mexican institutions where information was requested. Therefore, additional literature was considered in the weighting of the temperature and precipitation variables. For most studies and reports,

summer seasonality predominates [21, 60–63] indicate that seasonality indices showed two maximum peaks in summer. Although it has also been reported in May and June, due to the influence of changing habits during the summer period, with greater number of meals and outdoor recreational activities [64, 65]. Regarding the seasonality of *Campylobacter spp*. specifically in northern Mexico (Tamaulipas), it is reported that the incidence of the bacterium is influenced by contamination due to exposure to dust and environmental temperature during the summer months when the temperature reaches 38 to 40˚C [66, 67]. In countries with temperate climates, there is usually a higher incidence of cases in late spring and early summer [46, 68]. However, the cause of this phenomenon has not been clearly identified and different aspects have been related to seasonality. Other previous studies in Northern Europe show a significantly higher probability in summer [e.g.: 69–72], where higher ambient temperatures or with rainy days are influential [49, 73, 74]. Although there are studies that report a higher survival of the bacteria at cold temperatures [51, 52, 75]. Possibly the importance of some spatio-temporal risk factors that are going unnoticed may be underestimated [76].

In considering temperature, it was important to understand that *Campylobacter spp*. generally fall into two broad groups, catalase-positive and catalase-negative. The strains most often associated with human disease are catalase positive, these can be separated into two groups according to the temperature range in which they optimally thrive. Thermophiles have an optimal growth temperature of 42˚C and do not develop at 25˚C [14, 46]. In particular, *Campylobacter fetus* has received much attention for its importance in veterinary medicine, it is not thermophilic and cannot grow at 42˚C however, as the temperature range in modeling the potential occurrence in the study area of this group is typically diverse [77].

In the area of study, Mexicali (BC) has higher temperatures, reporting up to 43˚C in the warmest month, which could influence the presence of thermophilic species, and Santa Cruz (SON) is the municipality with the greatest temperate temperatures (add temperature range). Normally, most species grow optimally at 41.5˚C, although they can also grow at lower temperatures such as 37˚C, but not below 30˚C [11]. There are exceptions such as some strains of C. fetus, which are usually not able to grow at 42˚C but do grow at lower temperatures (even below 30˚C) [78]. Normally, the survival of *Campylobacter spp*. is favored at low temperatures, in the absence of light and at low oxygen concentrations, although the presence of competing organisms hinders their isolation [79]. Although *Campylobacter spp*. can also survive at cold temperatures [80]. Supported by studies showing that *Campylobacter* spp. can survive at refrigeration temperature in different biological environments for long periods, through adaptive responses to cold [10, 81, 82]. Studies have shown a correlation between environmental temperature and the number of cases. Several studies demonstrated that an elevated temperature prior to infection is the best predictor of cases in humans [e.g.: 83–85], with a negative peak during the summer, as well as an increase thereafter. This pattern is more in line with a bimodal distribution that is usually associated with the occurrence of outbreaks rather than the production of sporadic cases.

*Campylobacter spp*. is sensitive to harsh environmental conditions such as lack of humidity [11]. Therefore, in a dry environment survival is short [85, 86]. The risk associated with rainfall would be related to a higher survival of *Campylobacter spp*. in the air, as a consequence of the increase in relative humidity. In the European Union, it is not known if the increase in prevalence in summer is general, however, the environmental persistence of *Campylobacter spp*. could be very different in the south compared to the north. Precipitation in northwestern Mexico, specifically in the study area, is another limiting factor for the potential incidence of *Campylobacter spp*. because annual precipitation does not exceed 509.5 mm, which, in addition to the edaphic limitations, contributes to the fact that the municipality of Janos (CHIH) is the area with the highest potential incidence of this bacterium.

## Conclusion

The use of the kriging method of interpolation of unsampled areas is a tool that can help decision-makers to establish preventive and control strategies in high-risk areas and especially with infectious agents with complex cycles. Of the three municipalities evaluated, Janos (CHIH), has the highest number of geographic areas classified as high and medium incidence, followed by Santa Cruz (SON) and Mexicali (BC) with the highest number of geographic areas of low incidence zones.

In northern Mexico, known for its desert and extreme climatic conditions, the risk of the presence of *Campylobacter* spp. in the environment is mainly associated with agricultural areas with annual irrigation and low vegetation, regions that favor the survival and maintenance of *Campylobacter* spp. This, together with less extreme weather and soil conditions, mainly in the municipality of Janos (CHIH), favors humid environments with pH, salinity and soil texture conditions suitable for the maintenance of the bacteria.

In Mexico, records on the presence of *Campylobacter spp*. are scarce, even though it is the second most frequently reported foodborne disease, there is still much to be done in terms of research. Foodborne zoonoses are infections or diseases transmitted from animals to humans where it is essential to consider the integral cycle from water, soil, wildlife and their interaction with farm animals. A comprehensive approach to food safety from farm to fork is necessary to reduce campylobacteriosis. Soil, wastewater, wildlife and the various types of livestock, especially poultry, as well as farmers, industry, food inspectors, food vendors, food service workers and consumers are each a critical link in the food safety chain.

## Supporting information

**S1 Data.**
(XLSX)

## Acknowledgments

I would like to express my deepest appreciation to the members of the research team, for their invaluable contributions and collaborative spirit.

## Author Contributions

**Conceptualization:** Yasiri Flores, Andrea Chaves, Gerardo Suzán.

**Formal analysis:** Yasiri Flores.

**Funding acquisition:** Andrea Chaves, Gerardo Suzán.

**Methodology:** Yasiri Flores.

**Writing – original draft:** Andrea Chaves.

**Writing – review & editing:** Yasiri Flores, Gerardo Suzán.

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
