## [Decision Letter · Decision Letter 0]

23 Oct 2023

PGPH-D-23-01841

Prediction of edapho-climatic parameters in the incidence of *Campylobacter* spp. in northwestern Mexico

Dear Dr. Suzán,

Thank you for submitting your manuscript to PLOS Global Public Health. After careful consideration, we feel that it has merit but does not fully meet PLOS Global Public Health’s publication criteria as it currently stands. Therefore, we invite you to submit a revised version of the manuscript that addresses the points raised during the review process.

We look forward to receiving your revised manuscript.

Kind regards,

Ben Pascoe

Academic Editor

Journal Requirements:

1. In your Methods section, please provide additional information regarding the permits you obtained for the work. Please ensure you have included the full name of the authority that approved the field site access and, if no permits were required, a brief statement explaining why.

Additional Editor Comments (if provided):

Your manuscript has now been assessed by two independent reviewers who each found the topic of interest, particularly given the interesting use of modelling edaphic factors to assess the presence of *Campylobacter* in the environment. I would like you to revise the manuscript in line with their feedback below, specifically addressing concerns over the small number of records used within the model.

Reviewers' comments:

Reviewer's Responses to Questions

**Comments to the Author**

1. Does this manuscript meet PLOS Global Public Health’s publication criteria? Is the manuscript technically sound, and do the data support the conclusions? The manuscript must describe methodologically and ethically rigorous research with conclusions that are appropriately drawn based on the data presented.

Reviewer #1: Partly

Reviewer #2: Partly

2. Has the statistical analysis been performed appropriately and rigorously?

Reviewer #1: N/A

Reviewer #2: I don't know

3. Have the authors made all data underlying the findings in their manuscript fully available (please refer to the Data Availability Statement at the start of the manuscript PDF file)?

Reviewer #1: Yes

Reviewer #2: Yes

4. Is the manuscript presented in an intelligible fashion and written in standard English?

Reviewer #1: Yes

Reviewer #2: Yes

5. Review Comments to the Author

Reviewer #1: This manuscript is describing the link between environment and campylobacter spp, using a prediction model. I am a little confused how given you state that there is limited records and surveillance of Campylobacter in Mexico, there is no animal surveillance and so I am assuming your predictions are based on human cases, which of course not all will be reported and so I am concerned over the small number of records used within the model and so the links to Campylobacter may be not as concrete as the authors allude to. I question whether all of the figures are needed and if perhaps they could be reduced?

Minor comments:

L56 Is there any animal surveillance? Is campylobacter looked for in animal populations, even for research studies? Obviously, most environmental Campylobacter will be attributed to an animal, what species of campylobacter are found in humans in Mexico?

L85 Reference is needed for soil survival, I would expect some further details on how long Campylobacter spp. have been reported to survive in soil.

L153 Campylobacter spp. should be in italics

Results: Campylobacter data from the surveillance schemes, did they all use the same standardise culture methods/detection methods?

L269 What literature review? Do you mean you performed a review of the literature?

Reviewer #2: Summary: The manuscript describes an interesting approach of using mathematical modelling of geographic, environmental and geological (edaphic) factors to confirm and predict the presence of Campylobacter in the environment. They hope to improve surveillance by linking certain edaphic factors to higher incidence of human Campylobacteriosis.

Abstract

• The Abstract adequately summarises the study.

Introduction

• The Introduction gives a detailed background to the study, some sections are lengthy and would benefit from shortening, some sentences need rewriting (see details below).

• The aim is clearly stated.

Line 30 from not form

Line 53/54 Campylobacter is the primary bacterial agent, as far as I am aware there are viral causes that are more prevalent such as Rota- and Norovirus.

Line 54 I suggest “primary etiological agent” not first etiological agent

Line 55 The authors talk about WHO but reference the FDA, why not reference WHO?

Line 57 if children had diarrhoea then they weren’t asymptomatic?

Line 59/61 Campylobacter jejuni not jenuni

Line 60 “…in 40% of cases”, does this refer to Mexico or globally, please specify

Line 61 Does reference 7 belong to Line 58?

Line 70/71 suggested rewriting this sentence:” Soil, manure, aquatic environments and water are the natural niches where Campylobacter spp. are found and transmission to humans and susceptible animals can occur.” Or similar

Line 73/74 suggested rewrite:“The growth and survival of Campylobacter spp is related to external environmental factors such as water, heat, UV radiation and salinity.”

Line 85/98 Campylobacter in italics

Methods

• Over what time period where the various data points collected?

• The authors give a lot of detail about the kriging method, some of which might be better suited for the introduction or the discussion in conjunction with their findings.

• However, they do not explain the BIO designations used in the Figures and these are also not mentioned anywhere in the results.

Line 142 some information on the 19 variables that make up the profile would be interesting here.

Results

• The authors describe BIO variables in the Figure legends but this is not described the in the result text and it is difficult to see how they relate to the tables, could an extra column be introduced in Table 2 for example, or am I misunderstanding something?

• I do not feel that the Figure titles and legends give me enough information to understand the figures completely.

Line 201 References to ANUCLIM and Worldclim is needed here, particularly as the authors refer to them for the meaning of the colours in Figures 2-6, which are not given in a legend.

Line 214 Campylobacter spp., incidence

Discussion

• I struggle to understand how the authors put their findings into context with the literature. My expectations from the Introduction were that the model would describe the interplay of several factors to create ideal conditions for Campylobacter to be found but the authors seem to discuss individual factors such as humidity or temperature. I am obviously not an expert on this but I feel that more could have been made of the data?

• Overall, the lack of Campylobacter spp. data is mentioned several times, would a lack of data or differences in reporting between the areas skew the results of the model? If so, this needs to be discussed in more detail.

• The conclusions section is not a conclusion but more a discussion of limitations, I would suggest discussing the limitations and expand on the conclusions described in the Abstract at the end of the discussion section.

Line 220 Campylobacter is a facultative anaerobe, so I think this needs rephrasing for accuracy.

Line 222 the authors state that “recent studies have shown…” yet one of the papers referenced is from 2007

Line 223 Viable but non culturable state

Line 224-226 This sentence is hard to understand, do the authors mean maintenance instead of permanence for example? Please rephrase.

Line 236 C. jejuni

Line 238 the municipality

Line 251 Rollins et al is the original reference for the 120 days, not the papers sited, other papers show a shorter duration so this should be acknowledged.

Line 323/324 “Campylobacter spp. are the second most frequently reported cause of foodborne illness.” Where? In Mexico or globally? Also this is not the main conclusion of this data, see comment above.

6. PLOS authors have the option to publish the peer review history of their article (what does this mean?). If published, this will include your full peer review and any attached files.

**Do you want your identity to be public for this peer review?** For information about this choice, including consent withdrawal, please see our Privacy Policy.

Reviewer #1: No

Reviewer #2: No

---

## [Decision Letter · Decision Letter 1]

19 Mar 2024

Prediction of edapho-climatic parameters in the incidence of *Campylobacter* spp. in northwestern Mexico

PGPH-D-23-01841R1

Dear Dr Suzán,

We are pleased to inform you that your manuscript 'Prediction of edapho-climatic parameters in the incidence of *Campylobacter* spp. in northwestern Mexico' has been provisionally accepted for publication in PLOS Global Public Health.

Best regards,

Ben Pascoe

Academic Editor

Thank you for taking on the reviewer feedback and incorporating their suggestions into a revised version of the manuscript.

Reviewer Comments (if any, and for reference):

Reviewer's Responses to Questions

**Comments to the Author**

1. If the authors have adequately addressed your comments raised in a previous round of review and you feel that this manuscript is now acceptable for publication, you may indicate that here to bypass the “Comments to the Author” section, enter your conflict of interest statement in the “Confidential to Editor” section, and submit your "Accept" recommendation.

Reviewer #1: All comments have been addressed

Reviewer #2: All comments have been addressed

2. Does this manuscript meet PLOS Global Public Health’s publication criteria? Is the manuscript technically sound, and do the data support the conclusions? The manuscript must describe methodologically and ethically rigorous research with conclusions that are appropriately drawn based on the data presented.

Reviewer #1: Yes

Reviewer #2: Yes

3. Has the statistical analysis been performed appropriately and rigorously?

Reviewer #1: N/A

Reviewer #2: I don't know

4. Have the authors made all data underlying the findings in their manuscript fully available (please refer to the Data Availability Statement at the start of the manuscript PDF file)?

Reviewer #1: Yes

Reviewer #2: Yes

5. Is the manuscript presented in an intelligible fashion and written in standard English?

Reviewer #1: Yes

Reviewer #2: Yes

6. Review Comments to the Author

Reviewer #1: All comments have been fully addressed

Reviewer #2: Thank you for addressing all my review comments.

Some edits are needed for some sentences in the Discussion to make them easier to understand:

Lines 279/280/281: these sentences don't make sense together. Do you mean the fact that Campylobacter survives well in certain soils, makes theses particular soils more significant sources of human Campylobacter infections than other types of soil?

Lines 289/290/291: please rewrite the sentence

Lines 293/294/295: you seem to say the same thing twice here

As far as I understand, and please correct me if I am wrong, your current data cannot tell you if conditions contribute to the persistence or survival of Campylobacter, you can speculate on this but the data can only compare incidence of Campylobacter from public data with the environmental condition in the area at the time? So for example: if Campylobacter incidence is high in a certain area, your data cannot tell you for certain if the Campylobacter survived there because of e.g. the soil conditions or if Campylobacter is being re-introduced due to agricultural activities?

Therefore throughout the discussion, please take care with the use "persistence", "presence" and "survival", and do not use these interchangeably.

7. PLOS authors have the option to publish the peer review history of their article (what does this mean?). If published, this will include your full peer review and any attached files.

**Do you want your identity to be public for this peer review?** For information about this choice, including consent withdrawal, please see our Privacy Policy.

Reviewer #1: No

Reviewer #2: No
